# A Time-Domain Planning Method for Surface Rescue Process of Amphibious Aircraft for Medium/Distant Maritime Rescue

**Lu Yang** [1], **Rong Yin** [2], **Yuanbo Xue** [1], **Yongliang Tian** [1,*] and **Hu Liu** [1]

1   School of Aeronautic Science and Engineering, Beihang University, Beijing 100083, China
2   Aviation Industry Development Research Center of China, Beijing 100029, China
*   Correspondence: tianyongliang_buaa@163.com

**Abstract:** Medium/distant maritime rescue is significantly important in the development of maritime business. For typical medium/distant maritime rescue, the range limitation of helicopters and many difficulties between helicopter and ship cooperation lead to unsatisfactory rescue results. Compared to helicopters and ships, amphibious aircrafts could effectively solve the problems faced by helicopters and ships and meet the medium/distant maritime rescue demands with their long cruise range, high speed, high rescue capability and surface landing capability. Therefore, a time-domain planning method (TPM) based on the k-means* clustering algorithm and the genetic algorithm* is proposed in this study for the surface rescue process (SRP) of amphibious aircrafts in medium/distant maritime rescue. To simulate the SRP of amphibious aircrafts, an agent-based simulation environment of medium/distant maritime rescue was constructed based on the Python platform. Finally, a case study was carried out to verify its effectiveness and applicability. The results show that the TPM exhibits satisfactory rescue results for the SRP of the amphibious aircraft and that less than 1 h of delay time is recommended for the amphibious aircraft to rescue the persons in distress by using TPM.

**Keywords:** amphibious aircraft; surface rescue process; medium/distant maritime rescue; agent-based modeling and simulation





## 1. Introduction

Maritime rescue plays an important role in the development of the maritime business [1]. Maritime rescue can be divided into offshore rescue and medium/distant maritime rescue according to the distance from the coastline. At present, maritime rescue heavily relies on helicopters and ships [2–4]. There have been many studies, especially for offshore rescue [5–9]. However, as the scope of maritime business activities gradually expands to the medium/distant maritime regions [10], the probability of maritime accidents begins to increase in medium/distant maritime regions [11]. Research in the field of medium/distant maritime rescue is becoming increasingly important. However, there is insufficient relevant research on this specific field [11–13], of which most focus on the allocation and arrangement of the multiple rescue resources, such as rescue bases, rescue helicopters, and rescue ships. Nevertheless, compared to offshore rescue, the range limitation of helicopters and the many difficulties between helicopter and ship cooperation lead to unsatisfactory rescue results in medium/distant maritime rescue [14,15].

Compared to helicopters and ships, amphibious aircrafts can effectively solve the problems faced by helicopters and ships and meet the medium/distant maritime rescue demands because of their long cruise range, high speed, high rescue capability, and surface landing capability [16–19]. Therefore, this study on the amphibious aircraft applications for medium/distant maritime rescue is meaningful. Most of the current research on amphibious aircrafts has focused on the verification of a specific performance of the aircraft and exploratory design studies [20–24]. There is still a large gap in research on the applications of amphibious aircrafts. The main application areas of amphibious aircrafts are in

forest firefighting and maritime rescue [25]. For maritime rescue with amphibious aircrafts, Yang et al. proposed an optimal algorithm for searching routes of amphibious aircrafts, considering their flight characteristics and sea rescue capability [26]. Zhou analyzed the factors affecting the siting of amphibious aircrafts [27]. Du et al. established the amphibious aircraft selection decision model based on the improved fuzzy evaluation method [28]. Xiong et al. established a landing search and rescue risk assessment system for amphibious aircrafts. Based on the studied assessment system, they constructed the affiliation function of each index by the assignment method, and determined a fuzzy comprehensive evaluation process [29]. Wu not only used the k-means clustering algorithm, addressing the multi-amphibious aircraft rescue task assignment problem, but also used the ant colony algorithm for multi-aircraft rescue route planning [30]. However, there is no detailed study of the rescue process for amphibious aircrafts. Therefore, it is meaningful and necessary to carry out an in-depth study on the rescue process of amphibious aircrafts, which is the main motivation of this study. For medium/distant maritime rescue, one feasible way is the application of the mathematical methods for optimization. The application of the mathematical methods for optimization has been widely used in various fields, including transportation, military, economy, etc.

In terms of maritime rescue, mathematical methods have also been frequently utilized. Firstly, with the development of maritime weather observation technology, the drifting trajectory of the person in distress (PDT) can be obtained through more accurate environmental information by using mathematical methods, which can improve the efficiency and success of maritime rescue [31]. Then, for the maritime rescue process determination and optimization, Xiong et al. proposed a method for helicopter search area planning based on a minimum bounding rectangle and k-means clustering [32]. Liu et al. proposed an evaluation framework of helicopter maritime search and rescue which reveals the influence mechanism of uncertainty factors [33]. Pang et al. proposed a new target allocation method based on the improved genetic algorithm called the multi-string genetic algorithm [34]. Moreover, Ma et al. proposed an optimization model of emergency resources allocation, considering multiple restrictions, including accident black spots, the possible locations of rescue bases, different types of emergency resources, and rescue ships [35]. These studies show that mathematical methods have great applications in maritime rescue. Therefore, a surface rescue process (SRP) of amphibious aircrafts for medium/distant maritime rescue based on mathematical methods is proposed in this study. This method combines the adaptive k-means clustering algorithm (k-means*) with the optimal genetic algorithm (GA*). In addition, to improve the effectiveness and applicability of this method in practical situations, this study considers the relative motion of the amphibious aircraft and of the PDTs, differing from the traditional rescue methods which are based on stationary position [36,37]. Furthermore, the change in the PDTs' position and the change in the PDTs' health uncertainty are taken into consideration. In summary, a time-domain planning method (TPM) for the SRP of amphibious aircrafts in medium/distant maritime rescue, for the first time, is proposed in this study.

To simulate the SRP of amphibious aircrafts, one common method is to develop a simulation environment. Most existing research of simulation methods includes agent-based modeling and simulation (ABMS), discrete event system specification, and the system dynamic model [38–40]. The amphibious aircraft surface rescue simulation environment is constructed based on ABMS to solve the impact of the time-domain results on the simulation results in this study. The amphibious aircraft agent and the PDT agent are built to simulate the SRP of the amphibious aircraft for medium/distant maritime rescue.

The rest of this article is organized as follows: Section 2 introduces the TPM for SRP of the amphibious aircraft for medium/distant maritime rescue. Section 3 mainly describes the construction of the simulation environment based on ABMS. In Section 4, the rescue time and the successful rescue rates of the PDTs in a capsized ship are calculated based on the TPM for model verification. Finally, a summary of the article and the future work are given in Section 5.

## 2. Rescue Process of TPM for SRP

The SRP of amphibious aircrafts for medium/distant maritime rescue is shown in Figure 1. The probable drift trajectories of the PDTs should be predicted first. Next, the time when the amphibious aircraft arrives at the distress position should be calculated. In order to maximize the rescue performance of the amphibious aircraft, the PDTs should be clustered when the amphibious aircraft arrives. Additionally, the uncertainty health weight of individuals in each cluster should be calculated. The criterion of selecting a water landing site is based on the maximum uncertainty health weight in the clusters, and should be the destination of the amphibious aircraft. After the water landing site is selected, the lifeboat on the amphibious aircraft should be sent out to rescue the PDTs in turn. Therefore, it is necessary to optimize the rescue sequence of the PDTs in this cluster to save rescue time. Then, the amphibious aircraft should return to the base after reaching its maximum capacity, otherwise it will change the clusters. Finally, the rescue should be finished when the end condition is satisfied. Otherwise, the rescue should be implemented again by the amphibious aircraft.

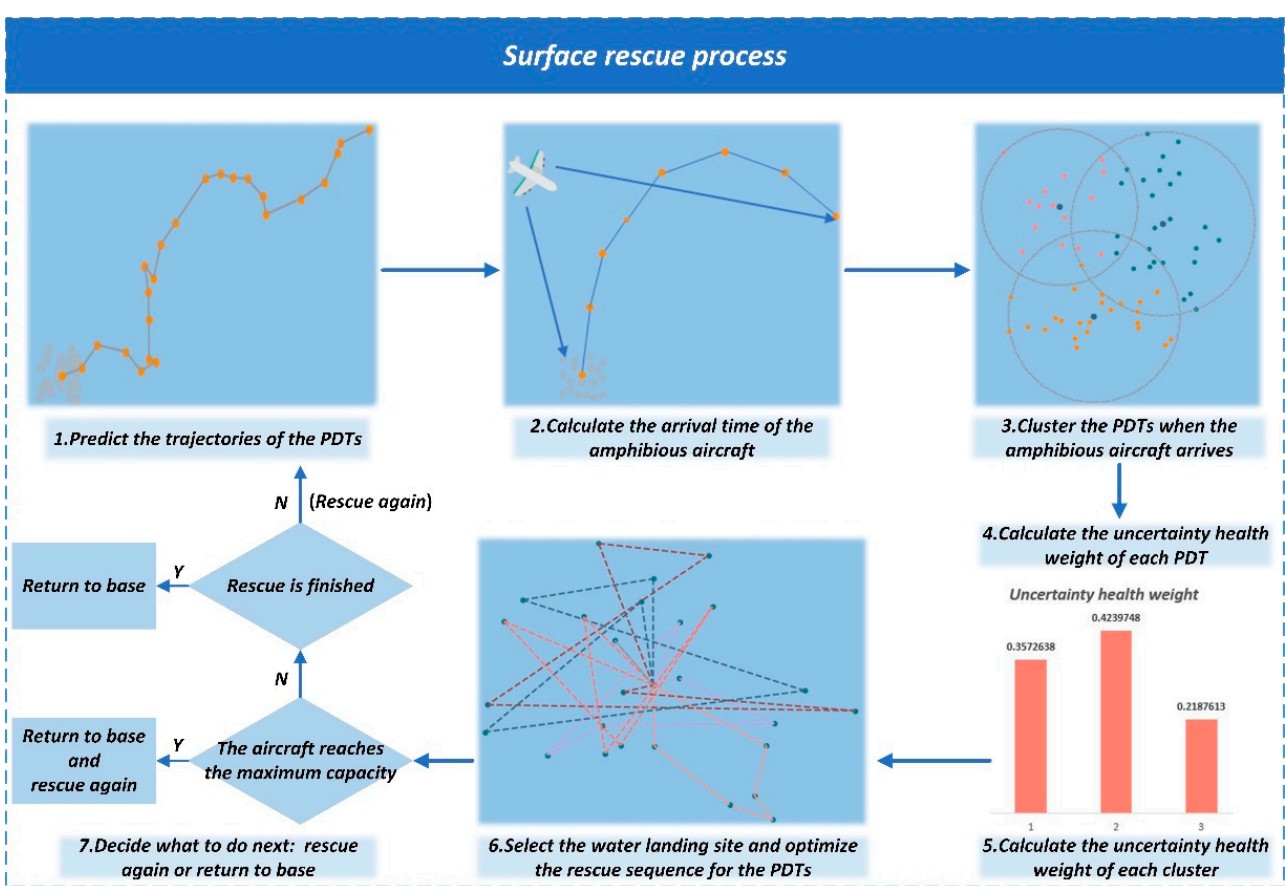

**Figure 1.** The surface rescue process of amphibious aircrafts based on TPM for medium/distant maritime rescue.

### 2.1. Predicting the Trajectories of PDTs

For medium/distant maritime rescue, it is worthy to predict the trajectories of the PDTs first. The drift trajectory can be influenced by winds, currents, and the physical properties of the PDT. Assuming that the ocean wind speed is $V_{wd}$, the ocean flow speed is $V_{wt}$, and the wind speed modification factor is $\gamma$. Then, for the maritime rescue mission, the drift speed of PDTs can be given by:

$$V_{drift} = V_{wt} + \gamma \cdot V_{wd} \tag{1}$$

Assuming that the initial position of the PDT is $s_0$ at a certain time, $t_0$, the ocean wind speed and the ocean flow speed do not change in a short time, $\Delta t$, and the position of the PDT after a short time can be predicted by:

$$s_{drift} = s_0 + \int_{t_0}^{t_0+\Delta t} V_{drift} dt \tag{2}$$

Considering that the ocean wind speed and the ocean flow speed, etc., are uncertain, and that the $\Delta t$ cannot be infinitely small, Monte Carlo experiments were used in this study which could simulate these uncertainties [41]. Additionally, in an actual rescue process, the lifeboat would have a certain range of rescue capabilities. Although the predicted location was not the accurate position, the PDTs could be rescued by the lifeboat within a little amount of time after it reached the predicted position of the PDTs. Therefore, the average predicted result of the Monte Carlo experiments was used to find the predicted trajectory of each PDT to improve the successful rescue rate. As shown in Figure 2, four examples of trajectory prediction were given.

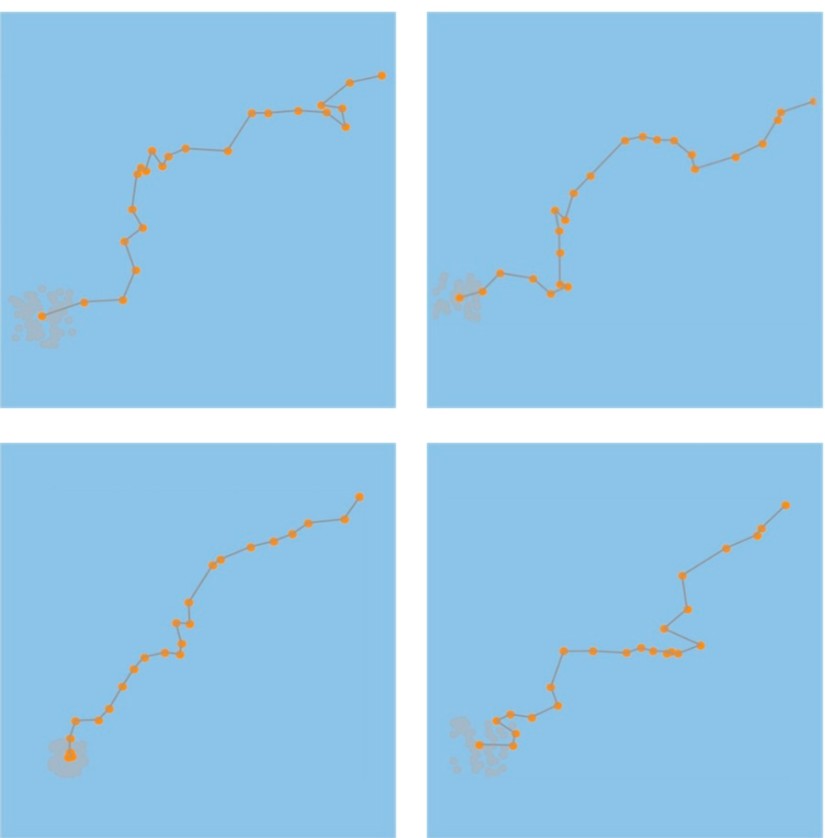

**Figure 2.** Four examples of average trajectory prediction over 24 h.

### 2.2. Calculating the Time to Reach the Distress Position

After obtaining the drift prediction trajectory of the PDTs, assuming that the distance between the rescue base and the distance position is $L$, and that the cruising speed of the amphibious aircraft is $V_c$, the time when the amphibious aircraft arrived at the distress position was calculated by:

$$T_{arrive} = \frac{L}{V_c} \tag{3}$$

### 2.3. Dividing the PDTs into Multiple Clusters by K-Means*

Assuming that the position of the PDTs at the initial moment, $T_0$, could be collected, the movement of the PDTs between $T_0$ and a short time, $\Delta T$, after $T_0$, was along a straight

line. Then, based on the arrival time of the amphibious aircraft, the drift position of the PDTs at $T_0 + T_{arrive}$ could be obtained with the initial position by Equations (1) and (2).

After determining the probable drift trajectory of the PDTs, the arrival time of the amphibious aircraft was calculated. Then, the lifeboat which was on the amphibious aircraft was sent out to rescue the PDTs after the amphibious aircraft landed. The lifeboat was used for rescuing and transferring the PDTs to the amphibious aircraft. Considering that the PDTs had different positions, and considering the limitation of rescue capability, it was necessary to determine where to land. In order to reduce the rescue time and increase the rescue performance of the amphibious aircraft, k-means* was used to divide the location of the PDTs in this study. Different from the typical k-means clustering algorithm [42–44], the k-means* clustering algorithm could adjust the optimal number of clustering through iteration based on the rescue capability of the amphibious aircraft as well as that of the lifeboat. Based on k-means*, the optimal number of clustered PDTs could be automatically obtained after iterative optimization.

The detailed optimization process of the k-means* clustering algorithm is shown as follows:

Step 1: give $k_{cluster}$ as follows. Based on the maximum rescue capability of the amphibious aircraft, $A_{num}$, the lifeboat voyage, $SL_{max}$, means the maximum distance it can travel without returning to the amphibious aircraft for resupply. The maximum rescue capability of the lifeboat, $S_{mn}$, means the maximum number of the PDTs carried, and the number of PDTs is $N_{input}$:

$$k_{cluster} = \left\lceil \frac{N_{input}[n]}{A_{num}} \right\rceil \quad (4)$$

Step 2: randomly select $k_{cluster}$ points as initial centers.

Step 3: calculate the distance to each center $k_{cluster}$ for each PDT in turn, and classify the PDTs to the nearest center $k_{cluster}$.

Step 4: calculate the new center of each cluster using the mean value method.

Step 5: if the position of the center changes, return to Step 3, otherwise, continue to Step 6.

Step 6: output the maximum distance $maxdis[cluster]$ of each PDT from the center of each cluster. If the maximum of $maxdis[cluster]$ is less than $SL_{max}/S_{mn}$, continue to Step 7, otherwise, assign $k_{cluster} + 1$ to and return to Step 2.

Step 7: output the position of the cluster centroids, $C[cluster]$, the number of the PDTs, $C_{num}$, and the position, $C_{location}$, for each category.

The flowchart of k-means is shown in Figure 3. As seen in Figure 3, the detailed process of k-means* is given. The pseudo code of k-means* is shown in Appendix A. In addition, Figure 4 shows the results of the clustering of the PDTs by k-means*. The number of the PDTs is 40 in Figure 4 (left) and 65 in Figure 4 (right).

### 2.4. Selecting the Water Landing Site Based on the Uncertainty Health Weight of PDTs

After obtaining $C[cluster]$, i.e., the location of the optional water landing site for the amphibious aircraft, it is important to choose the order of clustering points for the SRP of the multiple clustering centers. In this study, a method was proposed to calculate the uncertainty health weight based on the survival probability of the PDTs.

The survival time of the PDTs varied significantly because of the uncertainty of the health conditions of the PDTs. The maximum survival time of the PDTs after falling into the water can be given by:

$$T_{max}^i = \sigma_i \cdot 5.75 \cdot \exp(0.1 \cdot wt) \quad (5)$$

where $T_{max}^i$ is the maximum survival time of the ith PDT, $\sigma_i$ is the correction factor, whichdepends on the uncertainties of the PDTs, and $wt$ is the water temperature at the current position.

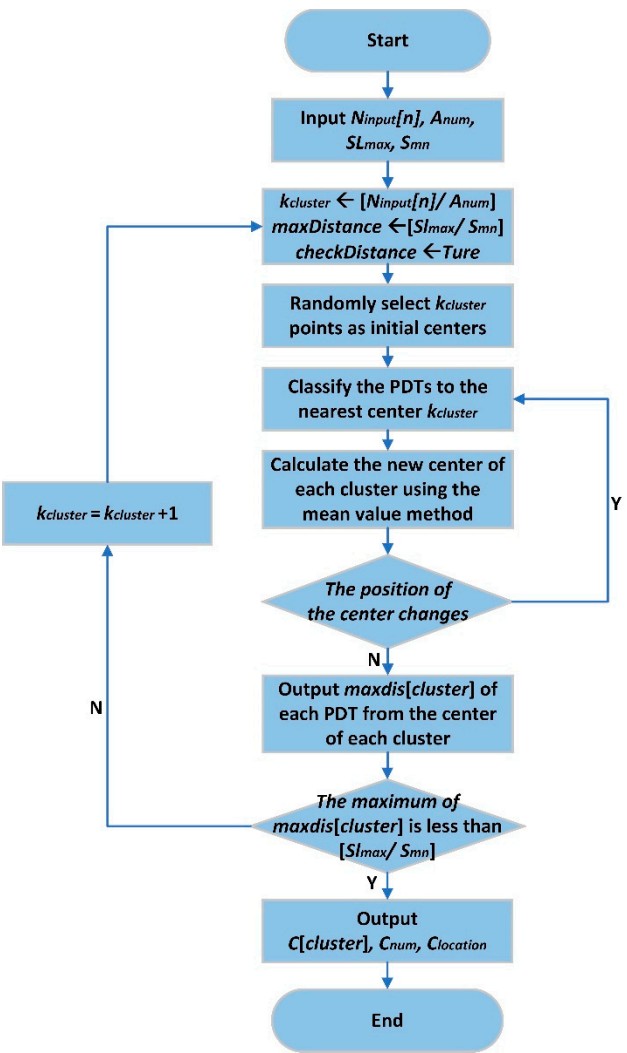

**Figure 3.** The flowchart of k-means*.

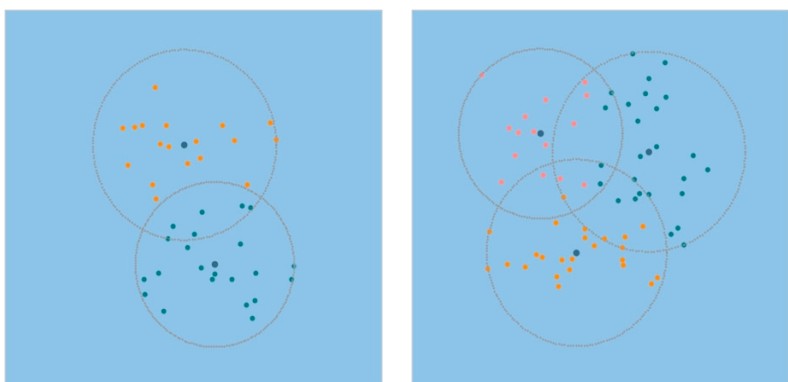

**Figure 4.** The results of clustering of the persons in distress by k-means*.

This study proposes the uncertainty health weight of the PDTs based on the maximum survival time. In this way, the survival probability of the i$^{th}$ PDT at the moment, $T$, after distress can be given by:

$$P_i = \begin{cases} \frac{T^i_{max} - T}{T^i_{max}}, & 0 \leq T < T^i_{max} \\ 0, & T^i_{max} \leq T \end{cases} \tag{6}$$

Next, the uncertainty health weight of each PDT can be given by:

$$W_i = \frac{P_i}{\sum_{i=0}^{Num-1} P_i}, \tag{7}$$

where $W_i$ is the uncertainty health weight of the $i^{th}$ PDT, and $Num$ is the total number of the PDTs.

Then, the uncertainty health weight in the $j^{th}$ cluster of the $C[cluster]$ clustering category result can be calculated as:

$$W^j = \sum_{i=0}^{C_{num}^j - 1} W_i^j \tag{8}$$

where $C_{num}^j$ is the number of the PDTs in the $j^{th}$ cluster of the $C[cluster]$ clustering category result, and $W_i^j$ is the uncertainty health weight of the $i^{th}$ PDT in the $j^{th}$ cluster. Finally, the optimal position can be chosen as the water landing site which is one with the maximum $W^j$.

### 2.5. Optimizing the Rescue Sequence for the Selected Cluster by GA*

After the amphibious aircraft selected the water landing site, the lifeboat was sent out to rescue the PDTs. One had to assume there was only one lifeboat in the amphibious aircraft. Furthermore, the positions of the PDTs were not going to change after selecting the water landing site. Since the positions of the PDTs would have been be differentiated, and since the lifeboat had a limited capacity, the lifeboat had to return to the amphibious aircraft when the maximum number of rescued persons was reached. Therefore, the issue at hand was how to choose the rescue sequence of the PDTs. In this study, the GA* was used to optimize the rescue sequence.

Different from the typical GA [45–47], in this study, the GA* could perform self-optimization by using LNS [48]. With LNS, the iterative optimization converges faster. Moreover, considering the limited rescue capability of the lifeboat, the optimization process of this study was slightly different from that of the typical GA. In detail, the lifeboat should have returned to the amphibious aircraft, which would have caused the rescue sequence to have many repeated points. Therefore, all operators were adjusted for better optimization. Furthermore, the detailed optimization process of the GA* is shown in the following section.

### 2.5.1. Population Definition

As mentioned in Section 2.3, the lifeboat was the only rescue responder for the PDTs. In a real rescue process, rescue results are usually influenced by the rescue sequence of the PDTs. For instance, it usually takes different amounts of time for rescue under different rescue paths. To include a consideration of this, the population was defined to describe the sequential paths of the lifeboat when rescuing the PDTs.

### 2.5.2. Individual Chromosome Coding

Assuming that the amphibious aircraft selected the $j^{th}$ cluster center as the water landing site, to highlight the characteristics of the population chromosome coding, the number of the PDTs in the cluster, $C_{num}^j$, is denoted by $N$.

For the $N$ PDTs of the cluster, it was digitally coded from 0 to $N-1$ as the chromosomal component of individuals within population. In addition, considering the limitations of the lifeboat rescue capacity mentioned previously, the water landing site of the amphibious aircraft was digitally coded as $N$. The chromosome code of a random individual in the population is shown in Figure 5 as an example.

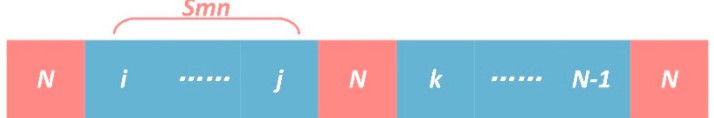

**Figure 5.** The chromosome code of the random individual in the population.

### 2.5.3. Optimization Objective

The optimization objective of the GA* was calculated with:

$$F = min \sum_{i=0}^{D} \frac{1}{d_{i(i+1)}} \tag{9}$$

where $D$ is the rescue sequence of the PDTs, $F(D)$ is the fitness function of the $D$, and $d_{i(i+1)}$ is the distance between the i$^{th}$ PDT and the $(i+1)^{th}$ PDT in the rescue sequence.

### 2.5.4. Self-Optimization Based on LNS

To improve the optimization ability of the GA*, in this study, individuals in the population were self-optimized based on LNS.

As shown in Figure 6, the studied LNS-based self-optimization algorithm was conducted as follows:

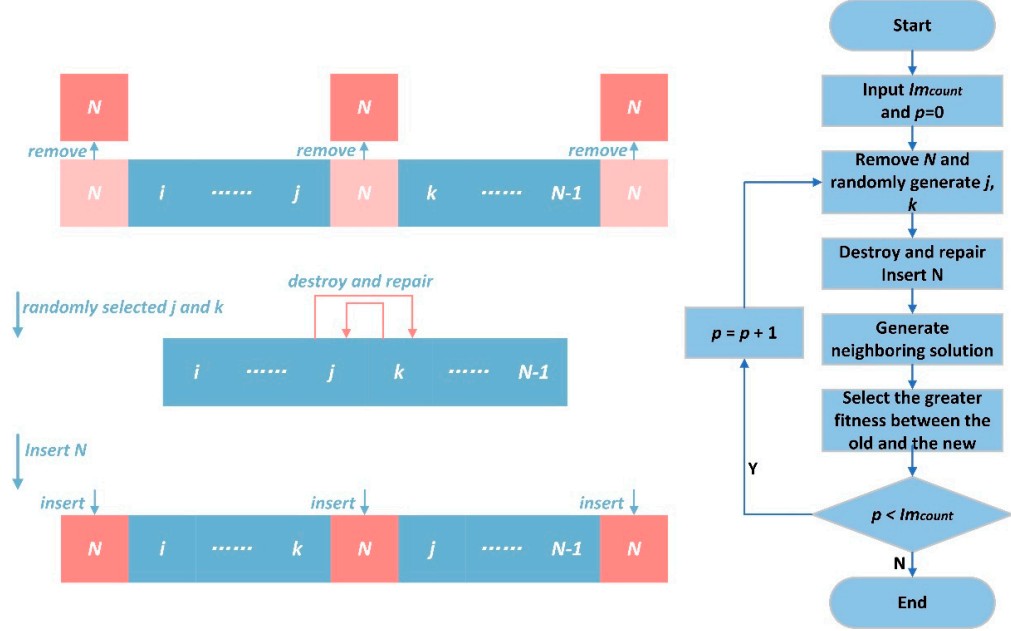

**Figure 6.** The schematic and the flowchart of self-optimization based on LNS.

Step 1: randomly select an individual in the population.

Step 2: determine the number of self-optimizations, $Im_{count}$, and initialize the number of the optimizations, $p = 0$.

Step 3: remove $N$ codes from the individual chromosome coding.

Step 4: randomly generate two integers $j, k$ to generate neighboring solution.

Step 5: destroy. Take out the code of the j$^{th}$ and k$^{th}$ position of the chromosome.

Step 6: repair. Put the j$^{th}$ code into the original k$^{th}$ position, and put the k$^{th}$ code into the original j$^{th}$ position.

Step 7: insert $N$ codes according to the rules mentioned in individual chromosome coding, then generate the neighboring solution.

Step 8: the fitness of the newly generated chromosome is compared with the fitness of the original chromosome, and the one with the greater fitness is taken as the new chromosome for the selected individual.

Step 9: if $p = Im_{count} - 1$, output the current chromosome individual, otherwise, use $p = p + 1$, and return to Step 2.

### 2.5.5. Selection Operator

In the selection operator, the elitist selection method was used. The elitist selection method was started by selecting a proportion of the most adapted individuals first in each generation of the selection process, and retaining the individuals in the next generation. The elitist selection method had a faster convergence of results and could find the global optimum faster. To prevent the population from falling into a local optimum as much as possible, the remaining individuals with a certain probability were randomly selected to be retained in the next generation.

### 2.5.6. Crossover Operator

A crossover operator was used to create new chromosomes from the retained individuals in the selection operator. This operator ensured the diversity of the population. As shown in Figure 7, the detailed process of using the crossover operator was as follows:

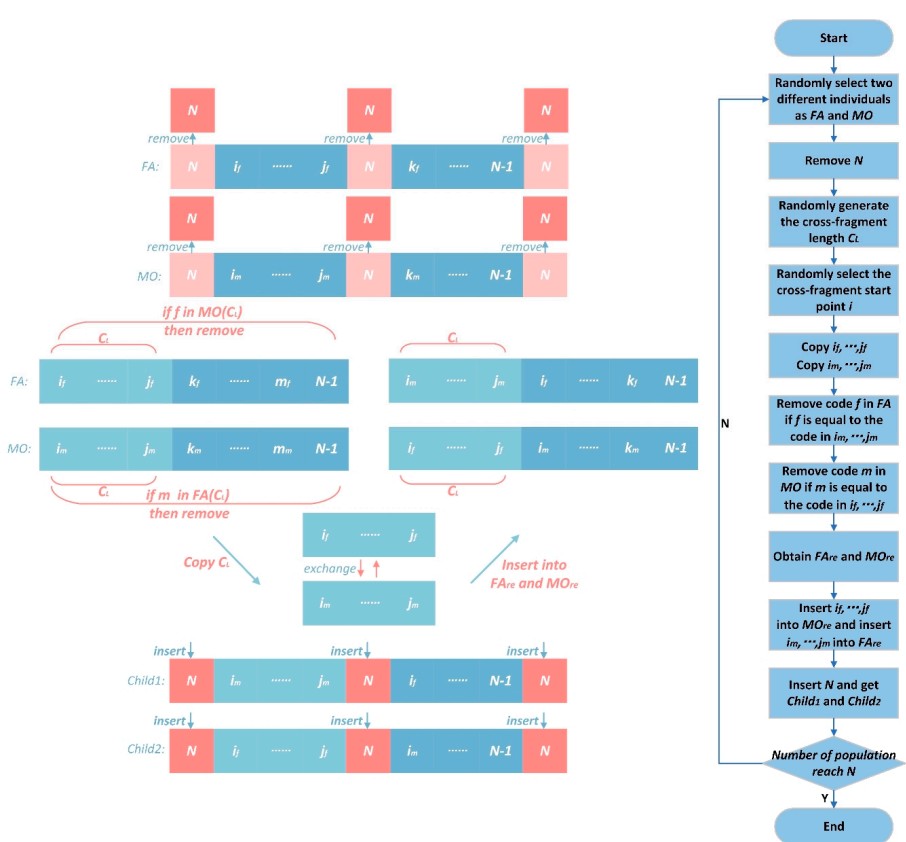

**Figure 7.** The schematics and the flowchart of the crossover operator.

Step 1: to satisfy population size stability, set the number of the individuals with the crossover operator *Child*, which is equal to the population size, $N$, minus the number of individuals retained by the selection operator.

Step 2: two different individuals retained by the selection operator are randomly selected as *FA* and *MO*.

Step 3: remove the $N$ codes from the individual chromosome coding, and denote the codes in *FA* and *MO* by $f$ and $m$, respectively.

Step 4: randomly generate the cross fragment length, $C_L$, and randomly select the cross fragment start point, $i$.

Step 5: copy the $i_f, \ldots, j_f$ and $i_m, \ldots, j_m$ fragments as cross fragments starting from $i$.

Step 6: remove the digitally coded $f$ in $FA$ if $f$ is equal to the code in the $i_m, \ldots, j_m$ fragments, and remove the digitally coded $m$ in $MO$ if $m$ is equal to the code in $i_m, \ldots, j_m$, to ensure that the codes cannot be duplicated. Then, $FA_{re}$ and $MO_{re}$ are obtained.

Step 7: insert the copied $i_f, \ldots, j_f$ fragments into $MO_{re}$, and insert the copied $i_m, \ldots, j_m$ fragments into $FA_{re}$.

Step 8: insert the $N$ codes according to the previously mentioned rules, then get the offspring $Child_1$ and $Child_2$.

Step 9: if the number of the population reaches $N$, stop the crossover operator, otherwise, return to Step 2.

### 2.5.7. Mutation Operator

To improve the exploration of the optimal solution, the mutation operator was used. The mutation operator involved a random selection of an individual in the $Child$ obtained with the crossover operator. The position of the two codes in the chromosome were randomly changed with a certain probability.

As shown in Figure 8, the detailed process of using the mutation operator was as follows:

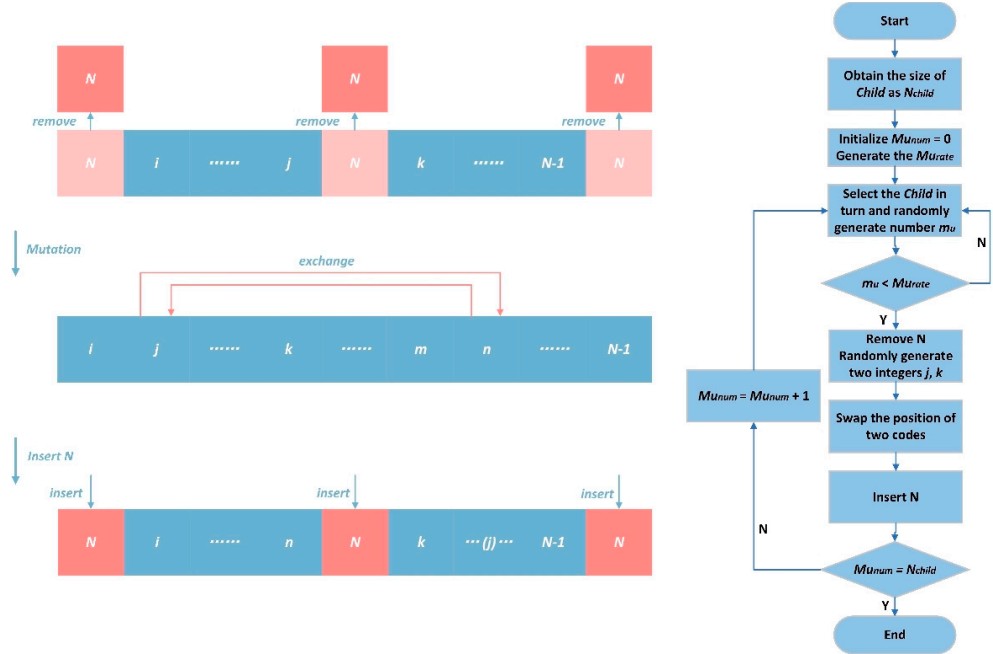

**Figure 8.** The schematics and the flowchart of the mutation operator.

Step 1: obtain the size of the $Child$ as $N_{child}$, then initialize the number of the mutation operators as $Mu_{num} = 0$.

Step 2: generate the $Mu_{rate}$, and perform Step 2 to Step 7 for each $Child$ obtained with the crossover operator.

Step 3: select the $Child$ in turn, and generate a random number, $m_u$. If $m_u$ is less than $Mu_{rate}$, continue to Step 4. Otherwise, perform Step 3 again and $Mu_{num} = Mu_{num} + 1$.

Step 4: remove the $N$ codes from the individual chromosome coding.

Step 5: randomly generate two integers, $j$, and $k$, and swap the position of the two codes.

Step 6: insert the $N$ codes according to the previously mentioned rules.

Step 7: if $Mu_{num} = N_{child}$, stop the mutation operator, otherwise, return to Step 3.

After selecting the water landing site, the population and the parameters of the population were initialized randomly. Then, within the evolution times, the population evolved by using the self-optimization based on LNS, selection operator, crossover operator, and mutation operator. The flowchart of the GA* is show in Figure 9.

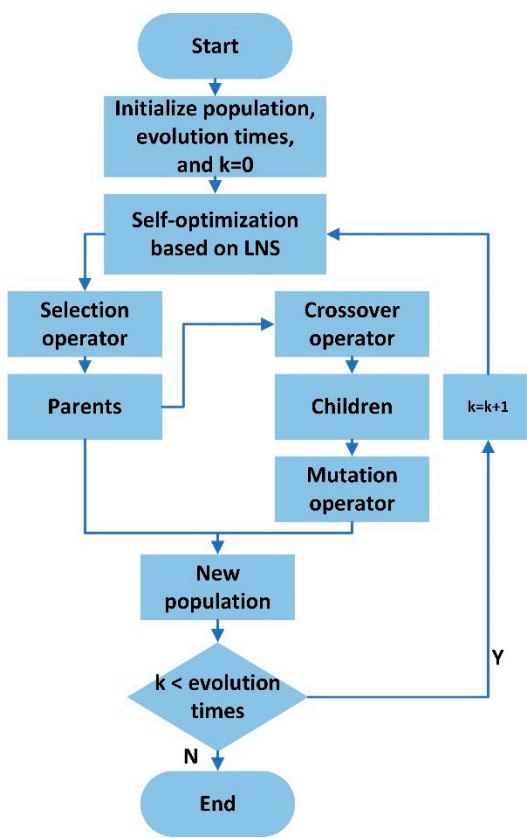

**Figure 9.** The flowchart of the GA*.

## 3. Simulation Environment

In this study, ABMS [36,49] was applied to construct the simulation environment by using the Python platform [50,51]. The amphibious aircraft and the PDT were built to simulate the SRP of the amphibious aircraft in medium/distant maritime rescue. In addition, the distress environment information needed to be set in the simulation environment by the user, this information including winds, currents, and water temperature.

In the simulation, the distress position and the airport position were set in the form of longitude and latitude by the user. Then, the distance between the distress position and the airport was calculated based on the longitude and latitude. The PDTs were placed at the initial location around the distress position. As time went on, the PDTs drifted along a certain trajectory, which was affected by the environment. In addition, the PDTs generated a maximum survival time according to their health at the initial time. After the amphibious aircraft received the rescue mission, the arrival time at the distress position was calculated. Then, the criterion of selecting the water landing site was based on the maximum uncertainty health weight in the clusters. Finally, the rescue of the PDTs was carried out.

### 3.1. The PDT Agent

In the simulation, the initial positions and the maximum survival times of the PDTs were generated at the initial time.

The behavior model of the PDT agent is shown in Figure 10. As time went on, the PDT agent departed from the initial position. The next drifting trajectory position of the PDT

was calculated by Equations (1) and (2), and the PDT agent moved to the next position after one simulation step in the simulation model. In this simulation environment, the simulation step length was one minute.

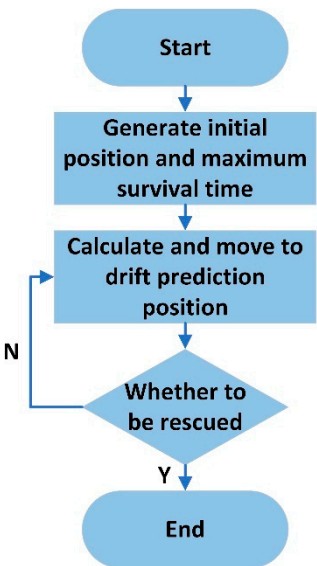

**Figure 10.** The behavior model of the PDT agent.

*3.2. The Amphibious Aircraft Agent*

In a simulation, the amphibious aircraft agent is generated at the rescue base with the performance parameters, including: cruising speed, the maximum rescue capacity of the amphibious aircraft, the number of carried lifeboats, the rescue capacity of the lifeboats, and the lifeboats' speed. Based on the cruise range and the refuel capacity at the rescue base of the amphibious aircraft, the fuel consumption for each rescue performed in this simulation environment was not considered.

The behavior model of the person in distress agent is shown in Figure 11.

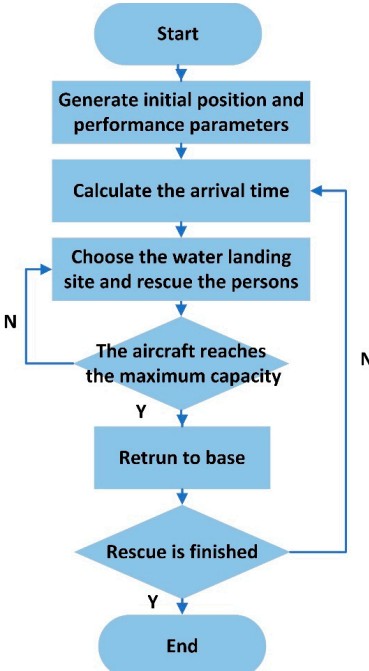

**Figure 11.** The behavior model of the amphibious aircraft agent.

Firstly, the arrival time at the distress position of the amphibious aircraft was calculated, and the drifting trajectories of the PDTs were predicted. The PDTs were clustered by k-means* based on the performance parameters of the amphibious aircraft. Then, the amphibious aircraft chose the water landing site, considering the uncertainty health weight of the PDTs. After the amphibious aircraft selected the water landing site, the lifeboat was sent out to rescue the PDTs following the optimized rescue path based on the GA*. Finally, after the rescue at the landing site was completed, it was be determined whether the amphibious aircraft should rescue again or finish the rescue according to the number of the rescued PDTs. If all the surviving PDTs were rescued, the rescue was finished. Otherwise, the amphibious aircraft was scheduled to rescue again. Additionally, the amphibious aircraft determined whether to continue the rescue or return to the base based on the number of PDTs on the amphibious aircraft.

## 4. Simulation Case and Analysis

Based on the simulation environment and the construction of the amphibious aircraft agent and the PDT agent, the effectiveness and applicability of the TPM is verified with cases in this section. In addition, the simulation environment could perform the TPM smoothly for different situations in the simulation case.

The simulation case was designed to be a capsized ship case that occurred in the distant sea region of the East China Sea. At 9:00 am on 11 June 2022, a ship capsized at the position 31°12′28.8″ N, 127°0′43.2″ E. The persons on the ship were all overboard. The average water temperature at the ship's position was 16 °C. In addition, the ocean wind speed and the ocean flow speed in the distress position could be obtained through weather monitoring applications.

Assuming that the number of PDTs was 45, the PDTs were randomly generated within a 2 km square around the distress position as the initial position. For each PDT, 50 samples were carried out in Monte Carlo experiments to predict their drifting trajectories based on the simulated time advance. Considering that the predicted trajectories were not the accurate positions, some extra time (5 min) was needed for the lifeboat to rescue one PDT.

The amphibious aircraft was located at Shanghai Pudong International Airport. The main performance parameters of the amphibious aircraft in the simulation environment are listed in Table 1.

**Table 1.** Performance parameters of the amphibious aircraft.

| Performance Parameters | Value |
|---|---|
| initial position | 31°9′0″ N, 121°48′21.6″ E |
| cruising speed (km/h) | 480 |
| aircraft guarantee time (refuel, transfer of the persons in distress, etc.) (min) | 30 |
| the maximum rescue capability (person) | 30 |
| number of lifeboats | 1 |
| lifeboat speed (km/h) | 28 |
| lifeboat voyage (km) | 92.6 |
| the maximum rescue capability of the lifeboat (person) | 5 |
| time to rescue one person of lifeboat (min/person) | 5 |

To the best of our knowledge, the distress information sent by the PDT cannot be immediately available to the rescue base in practical applications. Apart from that, the response and the preparation of the TPM will also take some time. Therefore, the start of the rescue may be delayed for some time while the PDTs are in distress. The TPM-X is used to indicate that the start of the rescue is delayed by X hours. Taking the TPM-0.5 as an example, the main screenshots of the TPM-0.5 and the simulation results for 45 PDTs are shown in Figure 12 and Table 2. To illustrate the TPM-0.5 of SRP in detail, the rescue time of each stage is shown in Figure 13.

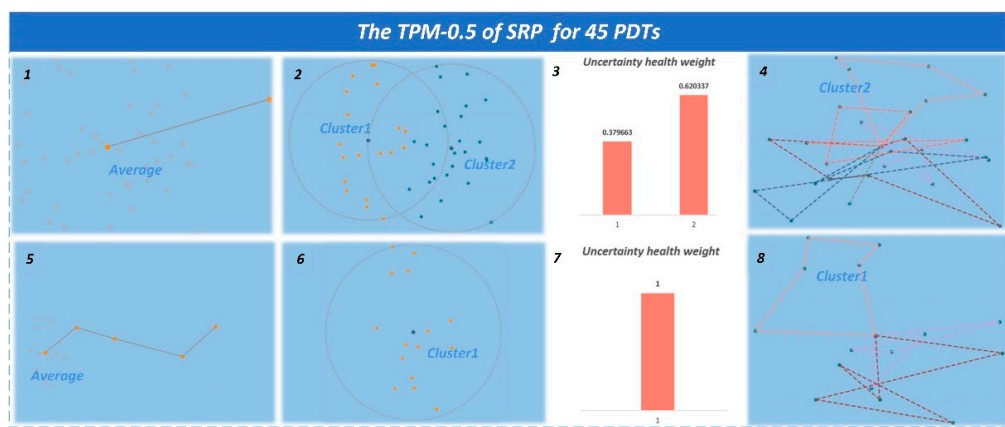

**Figure 12.** The TPM-0.5 of SRP for 45 PDTs.

**Table 2.** The simulation result for 45 PDTs of the TPM-0.5.

| Simulation Results | Value |
|---|---|
| Successful rescue rate | 91.1% |
| Rescue time (h) | 10.62 |

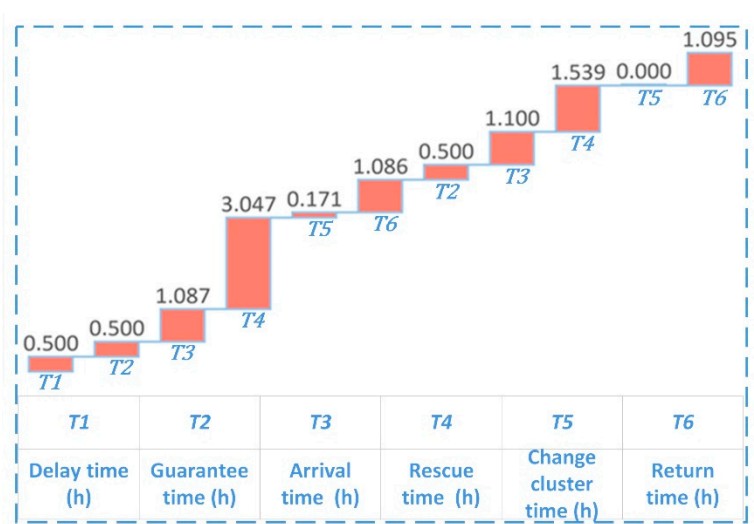

**Figure 13.** Rescue time of each stage for the TPM-0.5 with 45 PDTs.

The simulation results show that the TPM-0.5 had a high successful rescue rate and a short rescue time, which meant that the TPM-0.5 had greater effectiveness for 45 PDTs. The result shows that the amphibious aircraft needed to rescue the PTDs twice when there were 45 PDTs. The percentage of time taken to rescue the PDTs was the largest. This means it is necessary to optimize the SRP for medium/distant maritime rescue.

To verify the adaptability of the TPM for different rescue situations, a simulation experiment with 70 PDTs was conducted with other parameters equal to those mentioned in this study. Moreover, the initial positions of the PDTs were randomly generated, which caused uncertainties. The prediction trajectories of the PDTs also caused uncertainties in the Monte Carlo experiments, and the simulation results might be different for the same case because of uncertainty factors, including the hardware platform that supports the simulation environment. Therefore, to reduce the impact of random factors on the simulation results, the simulation of the TPM-0.5 for 45 and 70 PDTs was conducted fifty times separately. The box line diagram of the successful rescue rate and the rescue time for

the TPM-0.5 is shown in Figure 14. The results demonstrate the adaptivity of TPM with different numbers s of PDTs. The successful rescue rate decreases as the number of the PDTs increases, whereas the change in the rescue time follows the opposite trend.

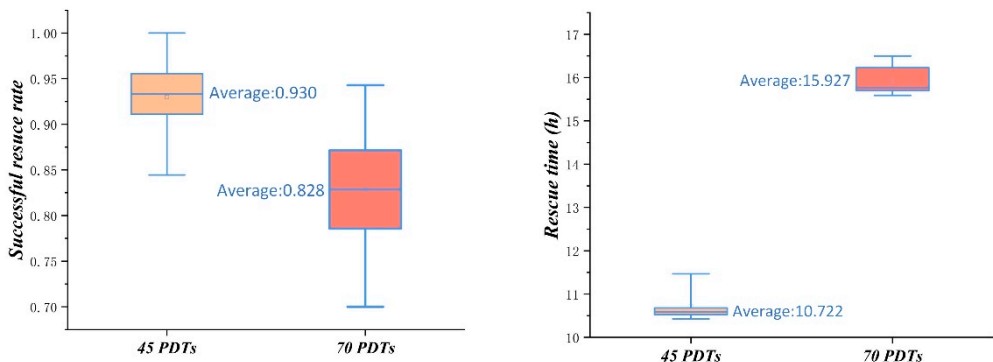

**Figure 14.** The box line diagram of the successful rescue rate and the rescue time for the TPM-0.5.

The average rescue times for each stage of the TPM-0.5 with 45 PDTs and 70 PDTs are shown in Figures 15 and 16. The meaning of $T_i$ in Figures 15 and 16 is the same as that in Figure 13. The results show that the number of the amphibious aircraft transfer sorties increased with the number of the PDTs due to the limitation of the amphibious aircraft rescue capabilities.

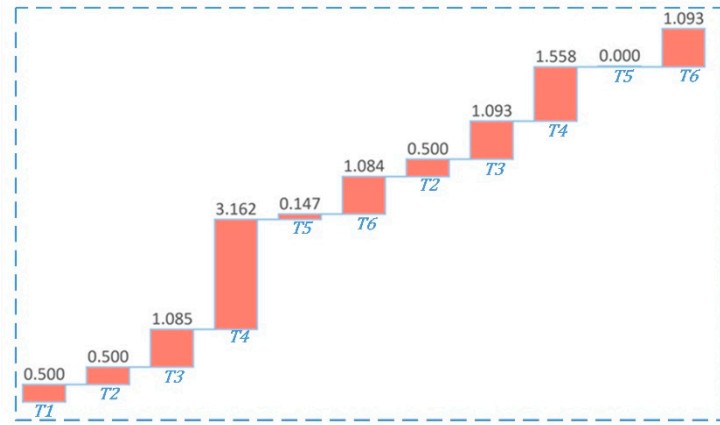

**Figure 15.** Average rescue time for each stage of the TPM-0.5 with 45 PDTs.

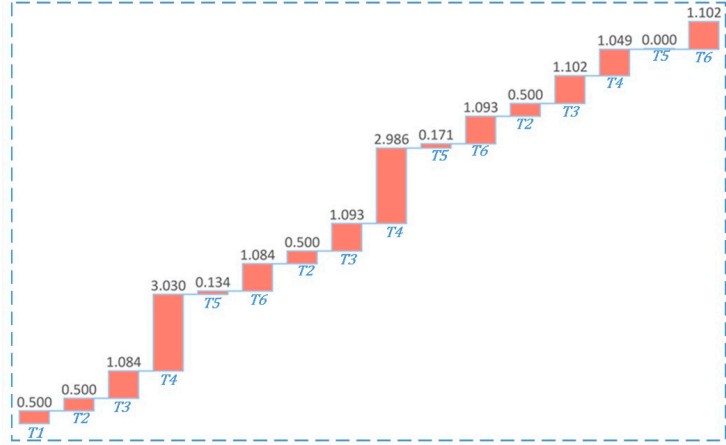

**Figure 16.** Average rescue time for each stage of the TPM-0.5 with 70 PDTs.

It can be seen that the percentage of time for rescuing the PDTs increased with the number of the PDTs. Therefore, the TPM has practical advantages for amphibious aircraft applications in medium/distant maritime rescue.

To investigate the effect of the time factor on the effectiveness of the TPM, TPM-X with a delay time varying from 1 h to 8 h was simulated for 45 and 70 PDTs, additionally. To reduce the impact of random factors on the simulation results, the simulation of each TPM-X was conducted fifty times. The simulation results of the successful rescue rate and the rescue time for 45 PDTs are shown in Figure 17. The successful rescue rate and the rescue time for 70 PDTs are shown in Figure 18.

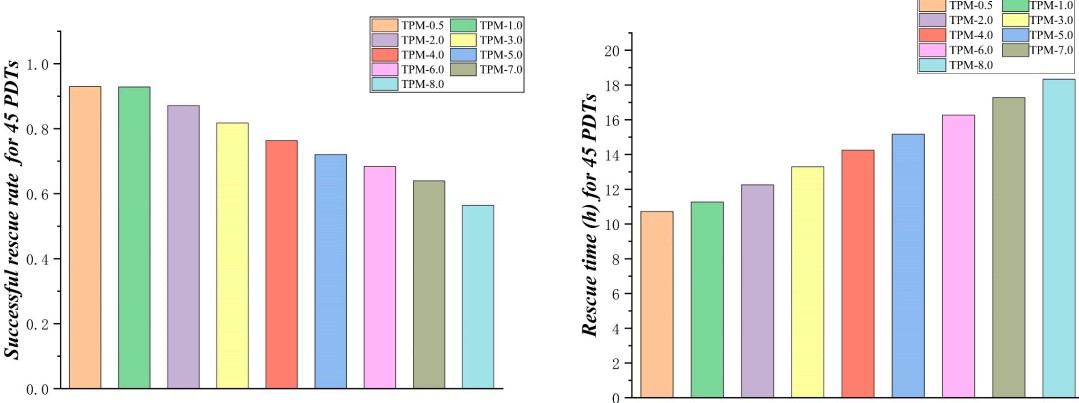

**Figure 17.** Successful rescue rate and rescue time for 45 PDTs with delay time from 0.5 h to 8.0 h.

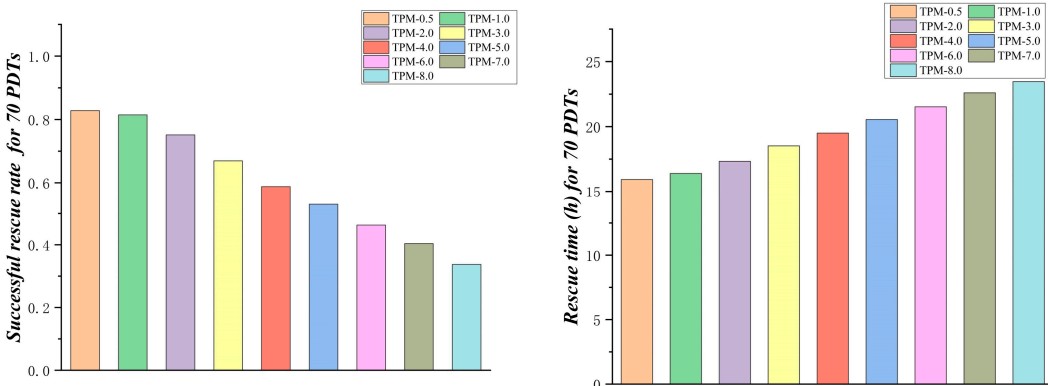

**Figure 18.** Successful rescue rate and rescue time for 70 PDTs with delay time from 0.5 h to 8.0 h.

It is noted that the TPM can significantly affect medium/distant maritime rescue. The effect of the time factor on the effectiveness of the TPM can be seen in the simulation results. The greater the delay time of the TPM, the lower the successful rescue rate and the longer the rescue time. The successful rescue rate for 70 PDTs decreased faster than that for 45 PDTs, with the delay time increasing. Moreover, the successful rescue rate and the rescue time of the TPM were affected by many factors, such as the drift speed of PDTs, the uncertain weather conditions, and the practical rescue experience of the amphibious aircraft. In addition, the accuracy of the trajectory prediction for the PDTs decreased over time.

Therefore, it is necessary to rapidly respond and prepare to the distress, especially when the number of the PDTs is high. Based on the simulation results of the TPM, for medium/distant maritime rescue, a delay time which is less than 1 h is usually recommended for the amphibious aircraft to rescue PDTs by using TPM.

## 5. Conclusions

Compared to offshore rescue, medium/distant maritime rescue is more challenging. However, most previous works have not fully developed a process for medium/distant maritime rescue, which is essential. The TPM is proposed to determine and optimize the SRP of the amphibious aircraft in medium/distant maritime rescue. To include the influence of the time variation, the TPM comprised the relative motion of the amphibious aircraft and of the PDTs, the change of PDTs' positions, and the changed uncertainty health weight of the PDTs. Based on this, the k-means* and the GA* were used to determine and optimize the SRP of the amphibious aircraft for TPM applications.

To evaluate the effectiveness and the applicability of the TPM, a simulation environment based on ABMS was constructed. The amphibious aircraft agent and the PDT agent were built to simulate the SRP of the amphibious aircraft in medium/distant maritime rescue. Finally, considering the practical rescue situation, it took a certain time delay for the rescue base to receive a request for rescue, respond to it, and prepare for the rescue. A capsized ship simulation with 45 PDTs and a 0.5-h time delay was designed to verify the effectiveness of the TPM. Moreover, to verify the applicability of the TPM for different situations, a simulation experiment with 70 PDTs and a 0.5-h time delay was additionally conducted with other parameters equal to those in this study. In addition, to investigate the effect of the time on the effectiveness of the TPM, a TPM with a delay time varying from 1 h to 8 h was simulated for 45 and 70 PDTs, respectively.

The simulation results show that the TPM has greatly applicability for different situations. In medium/distant maritime rescue, a delay time of less than 1 h is recommended for the amphibious aircraft to rescue PDTs using TPM. Correspondingly, it should be suggested that the rescue base responds and prepares quickly for rescue in practical situations. The proposed method proves to be available for effective rescue in a medium/distant maritime region. Moreover, the developed simulation environment can serve as a decision-making support tool in evaluating the effectiveness and the rapidity of medium/distant maritime rescue. To the best of our knowledge, the rescue results are unsatisfactory, as the number of PDTs increases because of the limitations on number of the amphibious aircraft and of the lifeboat in this study.

Finally, many extensions of this study could be considered in future work. One promising direction is the influence of other parameters on using TPM for medium/distant maritime rescue, such as lifeboat characteristics, the winds, the ocean flow, temperature, and other environment parameters. Moreover, the additional uncertainties of the PDTs could be considered, for example, the age, the weight, and the gender, instead of the correction factor ($\sigma_i$), for more accurate rescue results in an actual rescue process. This could not only retrieve more accurate rescue results, but could also enable the association between personal characteristics and simulation results to be observed. Furthermore, a multi-aircraft rescue method should be considered for more complex emergencies using amphibious aircrafts in medium/distant maritime rescue. This also provides a promising direction that allows a more accurate trajectory prediction for PDTs, since the accuracy of the trajectory prediction based on the Monte Carlo experiment decreased with a longer delay time.

**Author Contributions:** Conceptualization, L.Y. and Y.X.; Methodology, L.Y. and Y.X.; Validation, L.Y.; Data curation, L.Y.; Writing—original draft, L.Y. and Y.X.; Writing—review & editing, L.Y., R.Y., Y.X., Y.T. and H.L. All authors have read and agreed to the published version of the manuscript.

**Funding:** This research received no external funding.

**Institutional Review Board Statement:** Not applicable.

**Informed Consent Statement:** Not applicable.

**Data Availability Statement:** Not applicable.

**Conflicts of Interest:** The authors declare no conflict of interest.

**Appendix A**

```
1    input N_input[n], A_num, SL_max S_mn
2    k_cluster ← [N_input[n]/A_num]
3    maxDistance ← [SL_max/S_mn]
4    chenkDistance ← Ture
5    while chenkDistance
6         for N_k in N_input[n]
7              if i < k_cluster
8                   then Put N_k into C[k_cluster]
9         clusterChanged ← Ture
10        while clusterChanged
11             clusterChanged ← False
12             for N_i in N_input[n]
13                  minDist ← −∞
14                  minIndex ← −∞
15                  for C_j in C[cluster]  do
16                       if distance of N_i and C_j < minDist
17                            then minDist ← distance,
                                  minIndex ← C_j,
                                  Put N_i in to C^j_location
18                       if C_i ≠ minIndex
19                            then clusterChanged ← Ture
20             for C_j in C[cluster]  do
21                  Calculate the maxdistance_j of C_j and
                    all location that belong C^j_location
22                  Put maxdistance_j into maxdis[cluster]
23                  C^j_num ← number of C^j_location
24                  Put C^j_num into C_num
25                  Put C^j_location into C_location
26             if maxdistace of maxdis[cluster] < maxDistance
27                  then chenkDistance = False
28                  else k_cluster ← k_cluster + 1
29    return C[cluster], C_num, C_location
```

**Figure A1.** The pseudo code of k-means*.

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
