# Peer review of "A Time-Domain Planning Method for Surface Rescue Process of Amphibious Aircraft for Medium/Distant Maritime Rescue"

_applsci, doi:10.3390/app13042169_

Round 1
Reviewer 1 Report
The text fonts in Figs. 5-18 should be enlarged to be more visible and readable.
In my opinion, the subject and content of the manuscript are not suitable for this journal. I think another journal should be chosen for publication.
Reviewer 2 Report
In this work, the authors tried to develop a planning method based on k-means* clustering algorism for the process of medium/distant maritime rescue. The method was clearly described and the results were presented accurately. Thus, it is acceptable with minor revisions. The comments are shown below:
1. Please describe the usage of lifeboat and the relation between lifeboat and amphibious aircraft in line 150-151. The process of rescue should be described firstly and then explain the meaning of lifeboat voyage and capacity.
2. Please correct the grammar error in line 195.
3. What is the meaning of the sentence ‘Besides, it would be determined whether to rescue again or finish the rescue according to the number of the rescued PDTs.’ in line 343 and 344? Is the overall rescue process finished after all the PDTs rescued? Or will the process be stopped after a percentage of PDTs rescued?
4. Please provide the distance between the ship capsized location and the airport in the simulation mentioned in 3. Simulation Environment
5. Other than the delay time, is there any other parameters tested in the simulation? For example, the lifeboat characters or the ocean flow and temperature, which might affect the health weight of PDTs.
6. Please provide the simulation time and the hardware platform used to conduct this simulation. Is the simulation able to be finished within the aircraft delay time?
Reviewer 3 Report
This paper presents an investigation of time-domain planning method (TPM) for the surface rescue process (SRP). The results of the presented research are valuable to marine and rescue community.
However, before publication there are several aspects that should be addressed to provide readers with a better understanding of pertinent details used in this study and increase the value of the final conclusions:
1. The details of optimization used in this study must be clarified, also the average trajectory by Monte carlo.
2. Why the PDT has straight line, and not curved?
3. Please, mention the uncertainty of your simulation compared of case study.
4. The age, weight and gender of PDT are considered in this study?
Reviewer 4 Report
Dear Authors,
Experimental comparison is highly required to predict the behavior of nature and software value alone is not highly reliable for your research analysis. I have suggested to you for implementing in your research work for further improvement.
Data prediction-How many reliable values should be considered and implemented of y bar value in all figures based on your analysis data?
The reliable input data is necessary based on only optimization value but there is no evidence for any technique to identify the most influencing process parameters and contribution towards an improvement of the nature of the process.
Reviewer 5 Report
The paper is written well and I am going to accept the paper after considering the following comments:
1) I would suggest to change the title, in my view it is not clear. For instance the word “planning” is not specific.
2) In Figure 1, please mention every term such for example N, N*, Y**.
3) In section 2.3, please put the steps in the format of a flow chart.
4) It may be more appropriate to put the Figure 3 in the Appendix at the end of the paper.
5) Please add more explanation for the section 2.5.1.
6) Again, please put the steps in section 2.5.4 in the format of a flow chart as well as section 2.5.6, 2.5.7.
Round 2
Reviewer 1 Report
The improved manuscript may be published in the present form
Reviewer 4 Report
Dear Authors,
Experimental validation is one of the comparisons to predict the software values and also enhanced reliable prediction for your research work.